# The molecular clock protein Bmal1 regulates cell differentiation in mouse embryonic stem cells

Amador Gallardo[1,2,3], Aldara Molina[1,2,3], Helena G Asenjo[1,2,3], Jordi Martorell-Marugán[1,4], Rosa Montes[1], Verónica Ramos-Mejia[1], Antonio Sanchez-Pozo[1,2], Pedro Carmona-Sáez[1,6], Lourdes Lopez-Onieva[1,5], David Landeira[1,2,3]

**Mammals optimize their physiology to the light–dark cycle by synchronization of the master circadian clock in the brain with peripheral clocks in the rest of the tissues of the body. Circadian oscillations rely on a negative feedback loop exerted by the molecular clock that is composed by transcriptional activators Bmal1 and Clock, and their negative regulators Period and Cryptochrome. Components of the molecular clock are expressed during early development, but onset of robust circadian oscillations is only detected later during embryogenesis. Here, we have used naïve pluripotent mouse embryonic stem cells (mESCs) to study the role of Bmal1 during early development. We found that, compared to wild-type cells, Bmal1−/− mESCs express higher levels of Nanog protein and altered expression of pluripotency-associated signalling pathways. Importantly, Bmal1−/− mESCs display deficient multi-lineage cell differentiation capacity during the formation of teratomas and gastrula-like organoids. Overall, we reveal that Bmal1 regulates pluripotent cell differentiation and propose that the molecular clock is an hitherto unrecognized regulator of mammalian development.**

## Introduction

The earth rotates around its own axis with a 24-h period that generates repetitive changes in the intensity of sunlight reaching our planet. Organisms that live on the surface of the earth have developed mechanisms to optimize their physiology to this light–dark cycle. Circadian pacemakers allow to carry out an approximate measurement of time, so their phase must be adjusted daily to keep the internal clock in perfect synchrony with external signals. The main circadian synchronizer in mammals is light that is received in specialized photoreceptor cells in the retina, and the information is transmitted directly to the suprachiasmatic nucleus in the brain that synchronizes peripheral clocks through humoral signals such as hormones (Dibner et al, 2010). Most of cells of the adult organism have their own internal clock that needs to be synchronized to be in the same circadian phase as the rest of the body and, therefore, facilitate optimal physiological functioning (Mohawk et al, 2012). Circadian regulation relies on the activity of the molecular clock that mediates the establishment of an autoregulatory loop that generates daily oscillations in the expression of target genes (Takahashi, 2017). This machinery is composed by the core Clock and Bmal1 (also known as Arntl, aryl hydrocarbon receptor nuclear translocator-like) heterodimer that activates transcription of their own negative regulators Period (Per1, Per2, and Per3) and Cryptochrome (Cry1 and Cry2) genes. The molecular clock can regulate up to 10% of cellular transcripts in a tissue-specific way (Storch et al, 2002; Masri & Sassone-Corsi, 2010).

The function of the molecular clock during mammalian embryonic development is poorly understood (Seron-Ferre et al, 2012; Landgraf et al, 2014). Some components of the molecular clock are expressed during embryo development, but they do not generate consistent circadian fluctuations in embryo tissues until late stages of development when the suprachiasmatic nucleus is formed and the embryo is exposed to sunlight (Seron-Ferre et al, 2012; Landgraf et al, 2014; Umemura et al, 2017). In agreement, germline cells, zygotes, preimplantation embryos, and mouse embryonic stem cells (mESCs) derived from the developing blastocyst express components of the molecular clock but do not display circadian oscillations (Alvarez et al, 2003; Morse et al, 2003; Amano et al, 2009; Yagita et al, 2010). Importantly, despite mutant embryos lacking Bmal1 or other components of the molecular clock proceed through embryogenesis with no apparent phenotype at birth (van der Horst et al, 1999; Zheng et al, 2001; Kondratov et al, 2006; DeBruyne et al, 2007), recent evidence highlights that the lack of Bmal1 during embryo development is responsible for reduced life span, body weight, and fertility observed during the adult life in Bmal1−/− mice (Yang et al, 2016).

[1]Centre for Genomics and Oncological Research (GENYO), Granada, Spain  [2]Department of Biochemistry and Molecular Biology II, Faculty of Pharmacy, University of Granada, Granada, Spain  [3]Instituto de Investigación Biosanitaria, ibs.Granada, Hospital Virgen de las Nieves, Granada, Spain  [4]Atrys Health S.A., Barcelona, Spain  [5]Department of Biochemistry and Molecular Biology I, Faculty of Sciences, University of Granada, Granada, Spain  [6]Department of Statistics and Operational Research, University of Granada, Granada, Spain

Correspondence: davidlandeira@ugr.es

In this study, we have used naïve mESCs as a model system to analyse the molecular function of Bmal1 during early mouse development. We found that *Bmal1−/−* mESCs are viable and express pluripotency-associated markers. *Bmal1−/−* mESCs display increased expression of the pluripotency factor Nanog coupled to the mis-regulation of pluripotency signalling pathways. Importantly, the lack of Bmal1 protein in mESCs does not block pluripotency exit but perturbs the execution of multi-lineage cell differentiation as assayed using in vivo and in vitro cell differentiation systems. Our observations reveal a novel role of *Bmal1−/−* in the regulation of pluripotent cell differentiation and suggest that components of the molecular clock regulate mammalian development through molecular mechanisms that remain to be characterized.

## Results

### Bmal1 is expressed in pluripotent cells

mESCs do not show circadian oscillation of clock genes, but these can be induced by in vitro differentiation towards mouse neural stem cells (mNSCs) (Yagita et al, 2010). Reciprocally, circadian oscillations in mNSCs are extinguished when they are artificially reprogrammed to pluripotency (Yagita et al, 2010). We asked whether the onset of circadian oscillations during mESCs differentiation was associated with differences in Bmal1 expression levels. Colonies of mESCs growing in FBS and leukemia inhibitory factor (LIF) (mESCs FBS + LIF or primed serum mESCs) were differentiated into mNSCs that showed the typical fusiform shape morphology and grew as a reticular network (Fig 1A). mNSC cultures showed expected down-regulation of pluripotency genes *Nanog*, *Oct4*, and *Rex1* coupled to up-regulation of mNSC genes *Fabp7*, *Slc1a3*, and *Nestin* (Fig 1B). Immunofluorescence analysis confirmed that NSC cultures do not express the nuclear pluripotency-associated transcription factor Oct4 and showed homogeneous staining of the NSC protein marker Nestin in their cytoplasms (Fig 1C). Comparison of mRNA expression level in primed serum mESCs and NSCs showed that Bmal1 is expressed at a similar level in both cell types (Fig 1D). Expression of Bmal1 was similar to the transcriptionally active housekeeping gene *Hmbs*, indicating that Bmal1 is expressed at functional levels in mESCs (Fig 1D). Western blot analysis using anti-Bmal1 antibodies revealed two bands at 71 kD that correspond to previously described phosphorylated and unphosphorylated isoforms of Bmal1 (Yoshitane et al, 2009; Sahar et al, 2010) in mNSCs (Fig 1E). Importantly, primed serum mESCs and mNSCs expressed similar levels of Bmal1 protein (Figs 1E and S1A), suggesting that Bmal1 might be functional in pluripotent cells.

To discard that expression of Bmal1 protein was associated to marginal spontaneous cell differentiation and cell heterogeneity typically found in primed serum mESC cultures (Li & Izpisua Belmonte, 2018), we analysed expression of Bmal1 in naïve pluripotent cells. Primed serum mESCs were transferred to defined media with LIF and inhibitors of MAPK and glycogen synthase kinase 3 (2i + LIF media), to establish cultures of naïve 2i pluripotent cells (Ying et al, 2008). Expression of *Bmal1* was similar and slightly increased in naïve 2i mESCs as compared with primed serum mESCs, at the level of both mRNA and protein (Figs 1F and G and S1B), further indicating that Bmal1 is

expressed at a functional level in pluripotent cells. To confirm that expression of Bmal1 in pluripotent mESCs is not an artefact of in vitro cell culture conditions, we analysed whether *Bmal1* is also transcribed in pluripotent cells in vivo. We interrogated published single-cell messenger RNA high-throughput sequencing (mRNA-seq) datasets of developing preimplantation embryos (Deng et al, 2014) and found a heterogeneous pattern of *Bmal1* expression at all stages from four cells to late blastocyst (Fig 1H). Importantly, cells that express a higher level of the pluripotency gene *Oct4* also tend to express higher levels of *Bmal1* mRNA at the blastocyst stage (Fig 1I), demonstrating that *Bmal1* mRNA is expressed in pluripotent cells in vivo.

Last, cell fractionation followed by Western blot experiments showed that distribution of Bmal1 protein in mESCs is similar to its distribution in NSCs (nucleus and cytoplasm) (Figs 1J and S1C), suggesting that the absence of circadian oscillations in ESCs is not a consequence of the exclusion of Bmal1 from the nucleus of pluripotent cells. Taken together, these results solidly demonstrate that albeit mESCs do now display circadian oscillations (Yagita et al, 2010), the molecular clock protein Bmal1 is present in pluripotent cells.

### *Bmal1−/−* mESCs remain undifferentiated and express a higher level of Nanog protein

To investigate the role of Bmal1 protein in pluripotent cells, we derived *Bmal1−/−* mESCs using CRISPR/Cas9 and grew them in the 2i + LIF medium. The amino terminus of *Bmal1* was targeted (exon 5) (Fig 2A), and a panel of genetic clonal mESC lines were screened and genotyped. The frequency of *Bmal1* mutants in one or both alleles was high (four heterozygous [*Bmal1+/−*] and four compound heterozygous [*Bmal1−/−*] mutants of 12 screened clonal colonies), suggesting that Bmal1 is not essential to establish undifferentiated mESC cultures. Of the four *Bmal1−/−* clonal cell lines, we focused on one in which both alleles contained DNA indel mutations (−22 and +61 nucleotides, respectively) at expected target sites (Fig 2B) and resulted in frame shifts and premature stop codons at the very beginning of *Bmal1* open reading frame, impeding the production of peptides that include any of the annotated functional domains included in Bmal1 protein (BHLH, PAS1, PAS2, and TAD). *Bmal1−/−* mESCs formed typically tight three-dimensional colonies similar to the ones generated by parental wild-type cells (Fig 2C). As expected, expression of Bmal1 mRNA was not detected in *Bmal1−/−* mESCs compared with wild-type cells (Fig 2D), and Western blot analysis indicated that *Bmal1−/−* mESCs do not express detectable levels of Bmal1 protein (Fig 2E). *Bmal1−/−* mESC cultures remain undifferentiated because they retained expression of pluripotency-associated transcription factors *Oct4*, *Nanog*, and *Rex1* (Fig 2F). Interestingly, we found that *Bmal1−/−* mESCs express even higher levels of *Nanog* mRNA and protein than parental wild-type cells (Figs 2F–I and S1D). Taken together, these results demonstrate that Bmal1 protein is not essential for mESCs self-renewal, and that its depletion results in up-regulation of Nanog expression in naïve 2i mESCs.

### *Bmal1−/−* mESCs display altered expression of signalling pathway genes and cell differentiation potential

To further understand the role of Bmal1 in pluripotent cells, we compared mRNA expression in *Bmal1−/−* and wild-type naïve 2i mESCs by mRNA-seq. 854 gene transcripts were differentially

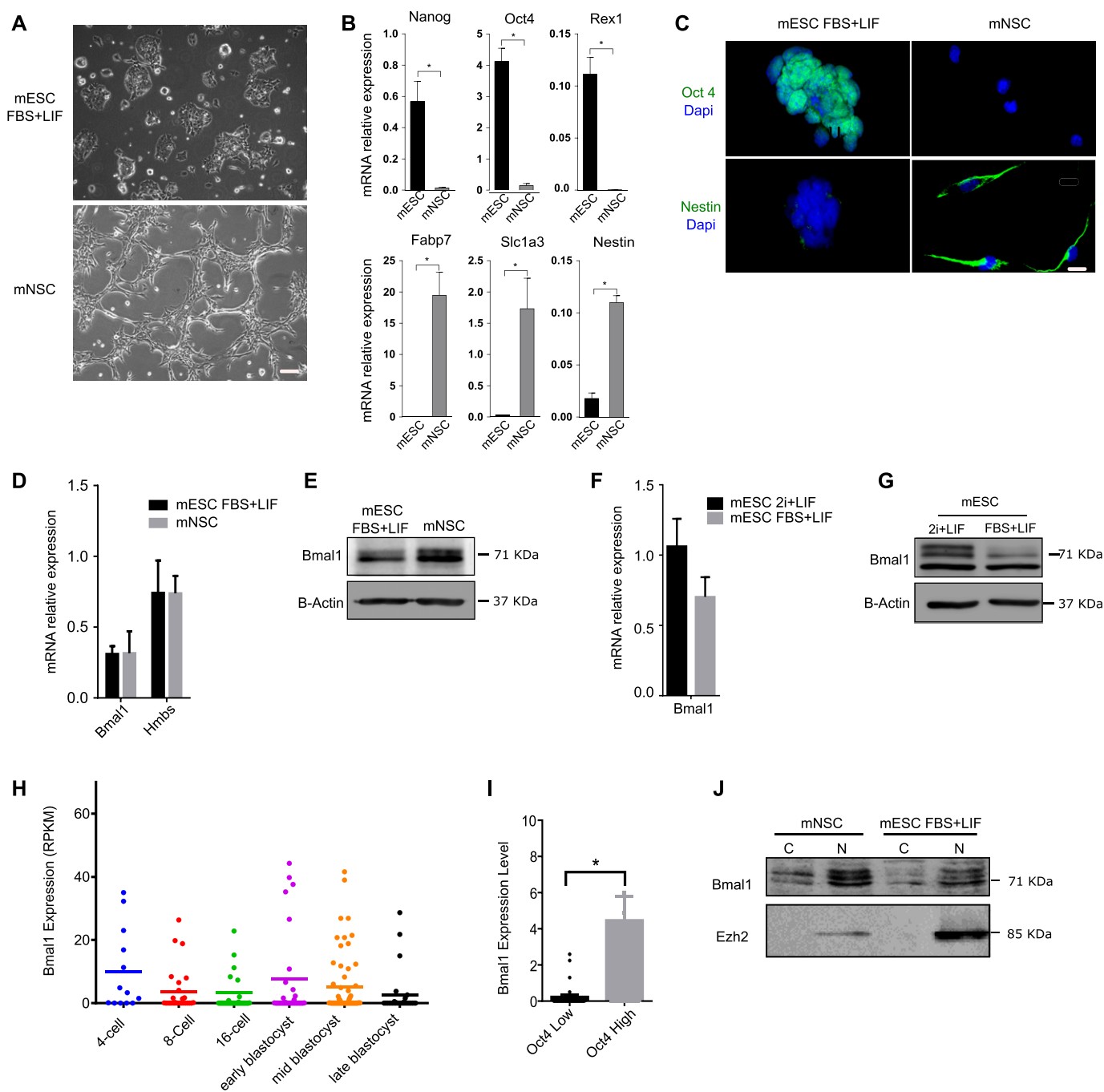

**Figure 1. Bmal1 is expressed in pluripotent cells.**
**(A)** Microscopic images of JM8 wild-type mouse embryonic stem cells (mESCs) and in vitro–derived mouse neural stem cell (mNSC) cultures. Scale bar is 100 μm. **(B)** RT-qPCR analysis of pluripotency-associated genes (*Nanog*, *Oct4*, and *Rex1*) and neural stem cell gene markers (*Fabp7*, *Slc1a3*, and *Nestin*) in mESCs and mNSCs. Expression is normalized to *Gapdh* and *RNA18s*. **(C)** Immunofluorescence images showing the expression of pluripotency (Oct4, green) and NSC (Nestin, green) protein markers in mESC and mNSC cultures. DAPI stains nuclear DNA (blue). Scale bar is 10 μm. **(D)** Expression of *Bmal1* in mESCs and NSCs measured by RT-qPCR. Expression of *Hmbs* was included as a control of a transcriptionally active and functional gene. Expression was normalized to *Gapdh* and *RNA18s* and multiplied by 10 to facilitate representation. **(E)** Western blot analysis of whole-cell extracts comparing Bmal1 protein levels in mESCs and mNSCs. B-actin provides a loading control. **(F)** RT-qPCR analysis comparing expression of *Bmal1* in mESCs grown in FBS + LIF or 2i + LIF media. Expression was normalized to *Gapdh* and *RNA18s* and multiplied by 10 to facilitate representation. **(G)** Western blot analysis of whole-cell extracts comparing Bmal1 protein levels in mESCs grown in FBS + LIF and 2i + LIF media. B-actin provides a loading control. **(H)** Single-cell mRNA expression values of *Bmal1* during indicated stages of preimplantation development calculated using previously published datasets (Deng et al, 2014). **(I)** Analysis of *Bmal1* expression in *Oct4* low and high individual cells during the blastocyst stage (early, mid, and late). Mean and SEM is shown. **(J)** Western blot analysis of Bmal1 protein localization in cell fractionation analysis of mESCs and mNSCs. Cytoplasmic (C) and nuclear (N) fractions are indicated. Ezh2 was used as a nuclear control. **(B, D, F)** Mean and SEM of three independent experiments are shown in (B, D, F). **(B, I)** Asterisks (*) in (B, I) indicate significant difference (t-Student $P < 0.05$).

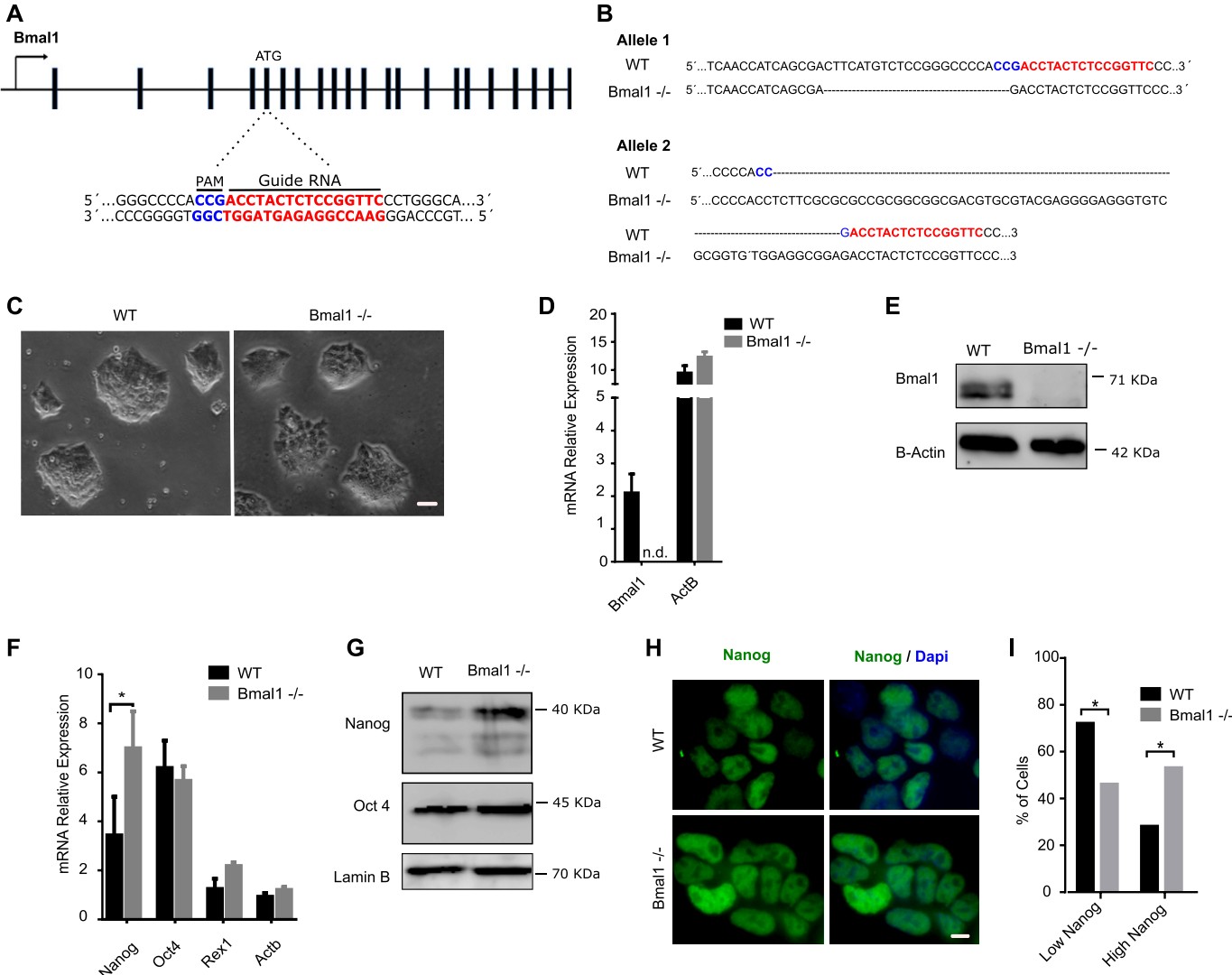

**Figure 2. Bmal1−/− mouse embryonic stem cells (mESCs) remain undifferentiated and express higher level of Nanog protein than wild-type cells.**
**(A)** Scheme of the *Bmal1* gene (black bars, exons) and the CRISPR/Cas9 strategy used. DNA sequence targeted by the guide RNA (red) and the localization of the PAM domain (blue) are indicated. **(B)** Alignment of DNA sequences of the *Bmal1* gene upon genotyping of *Bmal1−/−* mESCs obtained using CRISPR/Cas9. DNA regions targeted by the guide RNA are indicated in red. **(C)** Microscopy images of *Bmal1−/−* and wild-type mESCs grown in the 2i + LIF medium. Scale bar is 100 μm. **(D)** RT-qPCR analysis of *Bmal1* mRNA in wild-type and *Bmal1−/−* mESCs. *ActB* was used as a housekeeping control. Expression was normalized to *Gapdh* and *RNA18s* and multiplied by 10 for better representation. n.d. indicates non-detection of transcripts. Mean and SEM of three independent experiments is shown. **(E)** Western blot analysis of Bmal1 protein using whole-cell extracts of *Bmal1−/−* and wild-type mESCs. Lamin B provides a loading control. **(F)** mRNA expression analysis of *Nanog*, *Oct4*, and *Rex1* in *Bmal1−/−* and wild-type mESCs by RT-qPCR. Expression was normalized to *Gapdh* and *RNA18s* and multiplied by 10 for better representation. Asterisk (*) indicates significant difference (T-student $P < 0.05$). **(G)** Western blot analysis of whole-cell extracts comparing the protein levels of Nanog and Oct4 in *Bmal1−/−* and wild-type mESCs. Lamin B was used as a loading control. **(H)** Immunofluorescence analysis of Nanog protein (green) expression in *Bmal1−/−* and wild-type mESCs. DAPI stains nuclear DNA (blue). Scale bar is 10 μm. **(I)** Histogram showing the percentage of cells with low or high intensity of Nanog staining in parental and *Bmal1−/−* mESCs. Mean and SEM of three independent experiments is shown. Asterisks (*) indicate significant difference (Fisher's test, $P < 0.01$).

expressed (up- or down-regulated by more than twofold and adjusted *P*-value < 0.05) in cells that lack Bmal1 as compared with parental wild-type cells (Fig 3A). Unexpectedly, 689 of these transcripts (80.6%) were up-regulated in *Bmal1−/−* cells, suggesting that, in contrast to the role of Bmal1 in promoting transcription of target genes in adult tissues (Takahashi, 2017), Bmal1 is not acting as a transcriptional activator in pluripotent mESCs. Most of the up-regulated genes in *Bmal1−/−* mESCs were already transcriptionally active in wild-type cells (Fig 3B), suggesting that Bmal1 negatively modulates transcription of active genes. Functional categories

analysis revealed that *Bmal1−/−* mESCs display mis-regulation of genes encoding for 180 glycoproteins and 59 developmental proteins, suggesting that Bmal1 is involved in the regulation of cell communication and development (Fig 3C). In fitting, gene ontology analysis showed that depletion of Bmal1 results in deregulation of genes involved in cell differentiation and organism development (Fig 3D), and pathway enrichment analysis exposed that *Bmal1−/−* mESCs display altered regulation of pluripotency-associated signalling pathways including phosphoinositide 3-kinase (PI3K), MAPK, and Wnt (Fig 3E). In agreement with the general trend towards

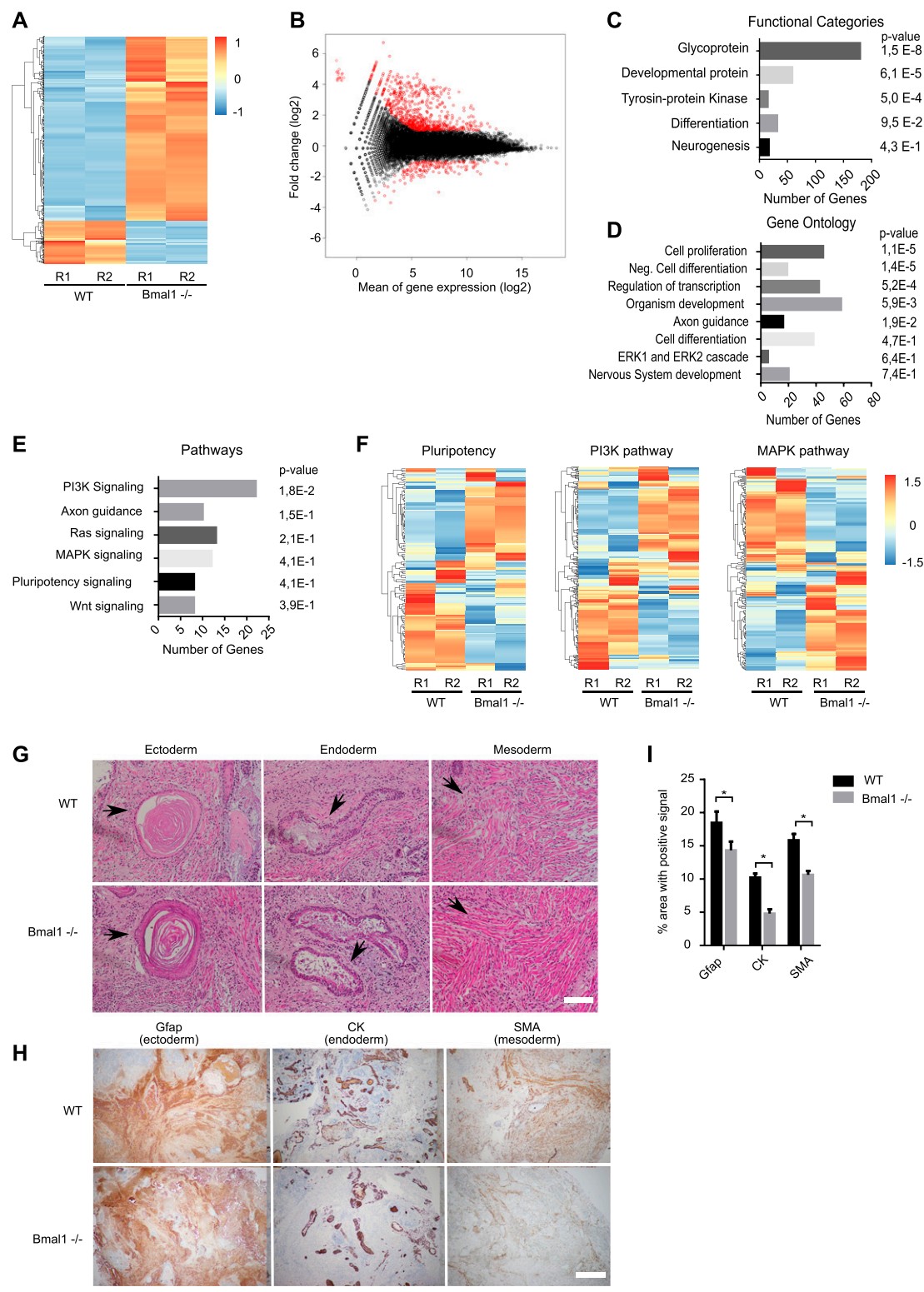

**Figure 3. *Bmal1−/−* mouse embryonic stem cells (mESCs) display altered expression of pluripotency signalling pathways and differentiation potential.**
**(A)** Gene expression analysis by mRNA-seq of *Bmal1-/-* and parental wild-type mESCs grown in the 2i + LIF medium using heat map clustering. Red and blue indicates higher and lower expression, respectively. Biological duplicates (R1 and R2) were carried out for indicated genotypes. **(B)** MA plot analysis of deregulated genes in *Bmal1-/-* mESCs. y-axis indicates fold change expression between *Bmal1-/-* and parental mESCs, whereas x-axis shows the mean expression level. Each dot represents one gene. Significantly mis-regulated genes are plotted in red. **(C, D, E)** Gene enrichment analyses of mis-regulated genes (854 genes with FC > 2 and $P < 0.05$) in *Bmal1-/-* mESCs. Categories are sorted by *P*-value and the number of genes included in each term are indicated in the x-axis. **(F)** Comparative expression analysis in *Bmal1-/-* and

up-regulation of gene transcripts found in *Bmal1*−/− mESCs, most differentially expressed genes involved in cell differentiation and embryo development were up-regulated in these cells (i.e., 47 of 58 genes involved in development, Fig S2A). In addition, analysis of components of signalling pathways deregulated in *Bmal1*−/− mESCs confirmed widespread abnormal expression patterns of mRNA transcripts involved in pluripotency signalling and cell differentiation including the MAPK pathway (Figs 3F and S2B). MAPK pathway, also known as extracellular signal-regulated kinases (Erk1/2) pathway, inhibits Nanog expression in mESCs (Wray et al, 2010), and thus, we wondered whether deregulation of this pathway underlie the up-regulation of Nanog protein found in *Bmal1*−/− mESCs (Fig 2F–I). However, expression of Erk1/2 protein was unaltered in *Bmal1*−/− mESCs in both primed serum and naïve 2i conditions (Fig S2C), and mRNA expression of Erk1/2 targets (Hamilton & Brickman, 2014) was not affected by depletion of Bmal1 protein (Fig S2D and E), suggesting that up-regulation of Nanog protein is not a consequence of reduced Erk1/2 signalling in *Bmal1*−/− mESCs.

We reasoned that up-regulation of Nanog protein and mis-regulation of signalling components might hinder the multi-lineage differentiation capacity of *Bmal1*−/− mESCs. We injected *Bmal1*−/− and wild-type mESCs into immunodeficient mice and assayed their ability to form teratomas and differentiate into the three germ layers. As expected, imaging analysis of teratomas using haematoxylin and eosin staining confirmed that wild-type mESCs can differentiate into tissues derived from the three germ layers. For example, we could detect skin (ectoderm), glandular tissue (endoderm), and smooth muscle (mesoderm) (Fig 3G). *Bmal1*−/− teratomas also contained tissues derived from the three germ layers, suggesting that the lack of Bmal1 does not fully block differentiation towards any the of three germ layers in vivo (Fig 3G). To complement this analysis using a quantitative approach, we performed immunohistochemistry using antibodies against glial fibrillary acidic protein (Gfap), cytokeratin, and smooth muscle actin that are well-established markers of ectoderm, endoderm, and mesoderm tissue respectively. Importantly, teratomas formed by *Bmal1*−/− cells displayed reduced labelling for Gfap, cytokeratin, and smooth muscle actin markers compared with wild-type cells (Fig 3H and I), indicating that although *Bmal1*−/− cells can produce tissue derived from the three germ layers, their differentiation potential is altered compared with wild-type cells.

We wondered whether reduced differentiation capability of *Bmal1*−/− mESCs was a consequence of abnormal expression of lineage specifier genes before teratoma formation. Analysis of mRNA expression of a subset of master regulators of ectoderm (*Sox1*, *Wnt7b*, *Pcdh17*, *Gsx1*, and *Pax6*), endoderm (*Gata4*, *Gata6*, *Foxa2*, *Cxcr4*, and *Sox7*), and mesoderm (*Bry*, *Bmp2*, *Hand1*, *Eomes*, *Mixl1*, and *Bmp4*) lineages showed that *Bmal1*−/− mESCs retain repression of lineage specifier genes (Fig S2F and G). This suggests

that differentiation defects found in *Bmal1*−/− teratomas are not a consequence of unscheduled transition of *Bmal1*−/− mESCs into a differentiated state, but rather mis-regulation of lineage specifier gene activation upon induction of cell differentiation.

## Bmal1 modulates multi-lineage differentiation in pluripotent cells

We compared the dynamics of lineage transition of *Bmal1*−/− and wild-type mESCs at early time points during cell differentiation using well-established in vitro systems. We induced cell differentiation by plating naïve 2i mESCs in N2B27 media growing as adherent cultures (Ying et al, 2003) and compared down-regulation of pluripotency genes and activation of differentiation markers by RT-qPCR in *Bmal1*−/− and parental wild-type cells (Fig 4A). As expected, wild-type cells down-regulated pluripotency genes (*Oct4* and *Nanog*; black bars) and induced expression of ectoderm (*Sox1*, *Wnt7b*, *Pcdh17*, *Gsx1*, and *Pax6*; blue bars), endoderm (*Gata4*, *Gata6*, *Bmp2*, *Cxcr4*, and *Sox7*; green bars), and mesoderm (*Bmp2*, *Hand1*, *Msx2*, *Eomes*, *Bmp4*, and *Mixl1*; red bars) genes after 4 d of differentiation (Figs 4B and C and S3A). Likewise, *Bmal1*−/− mESCs also down-regulated pluripotency genes and activated ectoderm genes (Figs 4B and C and S3A). However, activation of endoderm and mesoderm markers was severely hindered in differentiating *Bmal1*−/− cells (grey bars), as compared with wild-type control cells (Figs 4C and S3A).

To confirm that Bmal1 is required for activation of endoderm and mesoderm genes during pluripotent cell differentiation, we compared early time points upon pluripotency exit in primed serum *Bmal1*−/− and wild-type mESCs growing as adherent cultures (Fig 4D). *Bmal1*−/− mESCs down-regulated the pluripotency gene *Nanog* (Fig 4E) and induced expression of ectoderm markers (*Sox1*, *Gsx1*, and *Pax6*) with similar kinetics to wild-type mESCs (Fig 4F). Importantly, primed serum *Bmal1*−/− mESCs showed hindered activation of endoderm (*Gata4*, *Gata6*, *Foxa2*, *Cxcr4*, and *Sox7*) and mesoderm markers (*Bmp2*, *Hand1*, and *Msx2*) (Figs 4F and S3B). Taken together, these results indicate that Bmal1 is required to modulate multi-lineage differentiation in pluripotent mESCs and indicate that its activity favours endoderm and mesoderm specification.

## Bmal1 is required for the formation of gastrula-like organoids

To further examine the altered differentiation capacity of *Bmal1*−/− mESCs, we analysed their ability to differentiate and produce gastrula-like organoids (gastruloids) in vitro (Beccari et al, 2018). *Bmal1*−/− and wild-type mESCs were plated in low-attachment conditions to induce the formation of gastruloids, and their morphology and gene expression profile was compared during 5 d of differentiation. Compared with wild-type mESCs, *Bmal1*−/− cells formed smaller (average diameter on day 5 in wild-type is 371 ± 40 μm compared with *Bmal1*−/− 160 ± 16 μm) and less dense-looking

---

wild-type mESCs of genes involved in pluripotency, PI3K, and MAPK signalling pathways using heat map clustering. **(G)** Representative images of haematoxylin and eosin staining of teratomas derived from wild-type and *Bmal1*−/− mESCs. Tissues corresponding to three germ layers are shown (black arrow): skin for ectoderm, glandular tissue for endoderm, and smooth muscle for mesoderm. Scale bar is 100 μm. **(H)** Representative images of immunohistochemistry analysis of teratomas derived from wild-type and *Bmal1*−/− mESCs. Antibodies against Glial fibrillary acidic protein (ectoderm marker), cytokeratin (endoderm marker), and smooth muscle actin (mesoderm marker) were used. Scale bar is 500 μm. **(H, I)** Histogram showing quantification of the area of the images positive for indicated antibodies in teratoma immunohistochemistry analysis in (H). Mean and SEM of the analysis of 10 images are shown. Asterisk (*) indicates significant difference (t-Student < 0.05).

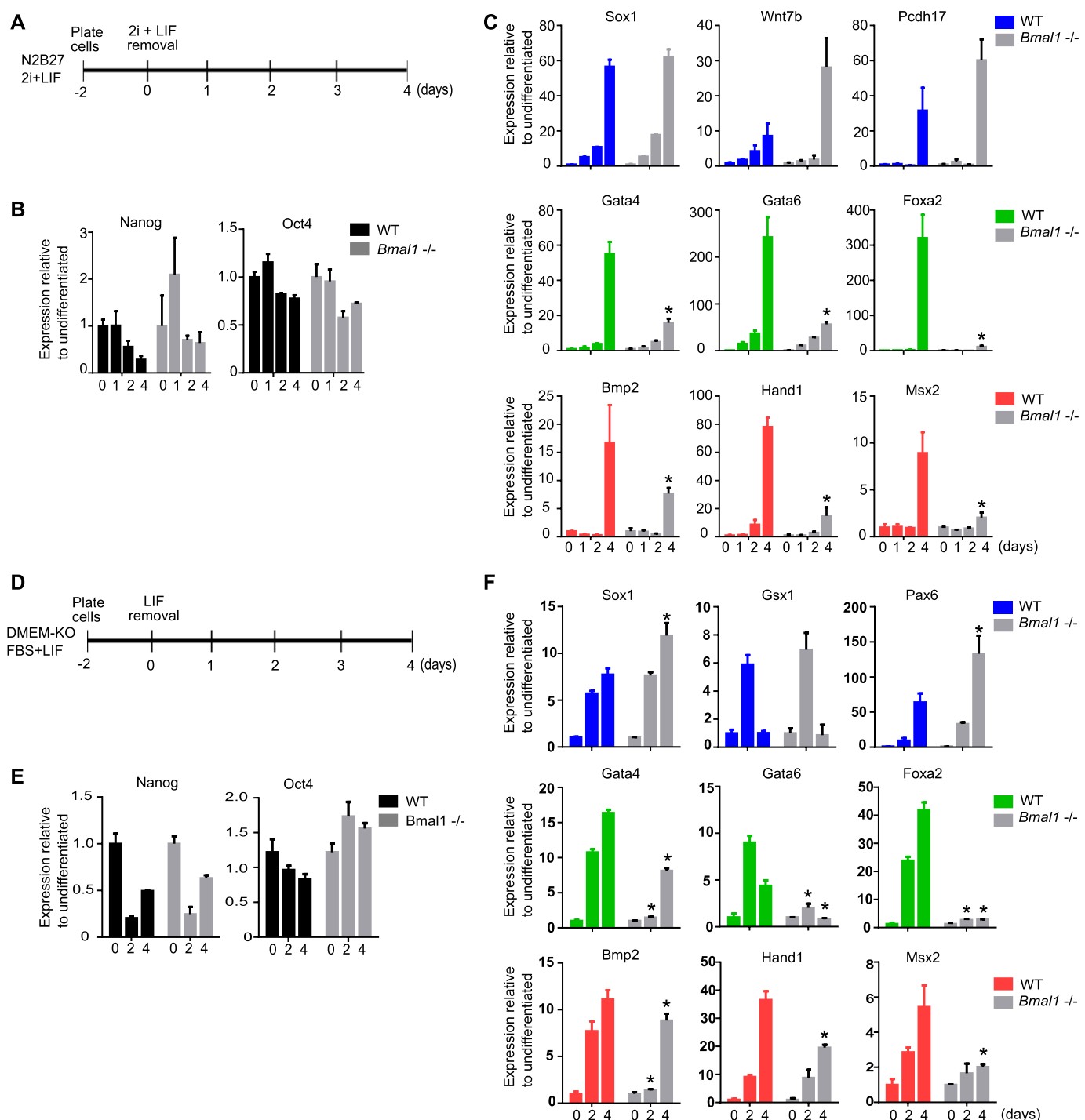

**Figure 4.** *Bmal1−/− mouse embryonic stem cells (mESCs) display reduced induction of endoderm and mesoderm genes during monolayer cell differentiation.* **(A, B, C)** Scheme of the protocol used to analyse differentiation of naïve pluripotent cells in (B, C). **(B, C)** mRNA expression of pluripotency (black), ectoderm (blue), endoderm (green), and mesoderm (red) specific genes, at indicated days during differentiation as measure by RT-qPCR in JM8 wild-type and *Bmal1−/−* (grey) mESCs. Expression was normalized to housekeeping genes *Hmbs* and *Ywhaz.* Fold change relative to time 0 is represented. **(D, E, F)** Scheme of the protocol used to analyse differentiation of primed serum pluripotent cells in (E, F). **(E, F)** mRNA expression of pluripotency (black)-, ectoderm (blue)-, endoderm (green)-, and mesoderm (red)-specific genes at indicated days during differentiation as a measure by RT-qPCR in JM8 wild-type and *Bmal1−/−* (grey) mESCs. Expression was normalized to housekeeping genes *Hmbs* and *Ywhaz.* Fold change relative to time 0 is represented. **(B, C, E, F)** Mean and SEM of three independent experiments are shown in (B, C, E, F). **(B, C, E, F)** Asterisks (*) in (B, C, E, F) indicate significant difference (T-student *P* < 0.05) between *Bmal1−/−* and JM8 parental cells at the same time point.

three-dimensional structures during the course of the experiment (Fig 5A, white arrows). Analysis of mRNA expression revealed that differentiating *Bmal1−/−* mESCs down-regulated pluripotency genes (*Oct4* and *Nanog*) and induced the activation of ectoderm genes (*Sox1*, *Gsx1*, *Wnt7*, and *Pax6*) at a similar or even more accused rate than parental control cells (Figs 5B and C and S3C). In contrast, *Bmal1−/−* differentiating gastruloids showed minor induction of endoderm (*Gata4*, *Gata6*, and *Foxa2*) and mesoderm (*Bmp2*, *Msx2*, and *Mixl1*) genes compared with parental control cells (Figs 5D and E and S3C). Thus, we concluded that *Bmal1−/−* mESCs consistently fail to activate endoderm and mesoderm genes upon pluripotency exit in all cell differentiation systems tested.

To confirm that the phenotype of *Bmal1−/−* mESCs is a consequence of Bmal1 protein depletion and is not linked to a CRISPR/Cas9 off-target effect, we derived an independent clonal population of *Bmal1−/−* mESCs (clone #G10). Genotyping confirmed that these cells are homozygous mutant for a deletion of four nucleotides in exon 5 in both alleles of the *Bmal1* gene that result in a frameshift and premature stop codon at the beginning of the open reading frame (Fig S4A). *Bmal1−/−* (clone #G10) did not express detectable levels of Bmal1 protein (Fig 5F), and similarly to the previously characterized *Bmal1−/−* mutant (Fig 2), *Bmal1−/−* (clone #G10) grown in the 2i + LIF medium formed typical tight three-dimensional colonies (Fig S4B) and expressed normal, or slightly increased, levels of the pluripotency-associated transcription factors *Nanog*, *Oct4*, and *Rex1* (Fig 5G). Importantly, *Bmal1−/−* (clone #G10) mESCs also displayed higher levels of Nanog protein than wild-type parental controls (Fig 5H) phenocopying the previous *Bmal1−/−* cells. In addition, analysis of the differentiation potential of *Bmal1−/−* (clone #G10) further confirmed that the lack of Bmal1 protein results in the formation of smaller gastruloids (Figs 5I and S4C) that can down-regulate pluripotency markers and up-regulate ectoderm genes but have compromised induction of endoderm and mesoderm markers (Figs 5J and S4D). Taken together, these analyses confirm that Bmal1 regulates Nanog expression and the multilineage potential of pluripotent cells and suggest that Bmal1 is required for formation of endoderm and mesoderm tissue during embryo development.

# Discussion

The most important discovery in this study is that the molecular clock protein Bmal1 is required for correct multi-lineage cell differentiation in pluripotent cells. This finding is interesting for the fields of circadian biology and pluripotency. It is currently not clear whether circadian proteins are expressed or not in pluripotent cells (Lu et al, 2016; Umemura et al, 2017), and therefore, our study is important to establish that the core component of the molecular clock Bmal1 is expressed in mESCs and that it has a functional role. Complementarily, our study show that circadian proteins are novel regulators of pluripotent biology and cell differentiation.

We show that Bmal1 regulates cell differentiation in mESCs where circadian rhythms cannot be detected using standard approaches (Kowalska et al, 2010; Yagita et al, 2010). Thus, our study support that during early development, the molecular clock carries

out a novel and uncharacterized molecular function that does not rely in the canonical oscillatory production of gene transcripts. Therefore, an important question that arises from our study is what is the molecular function of Bmal1 in mESCs and whether it is carried out in coordination with the rest of the machinery of the molecular clock. Although there is evidence supporting the absence of circadian oscillations in pluripotent cells (Kowalska et al, 2010; Yagita et al, 2010), it is not clear whether the lack of circadian rhythms is a consequence of the absence of Clock protein; conflicting reports show that Clock protein is present (Lu et al, 2016) or degraded by posttranslational mechanisms (Umemura et al, 2017) in mESCs. Although this study cannot solve this discrepancy, our results indicate that the function of Bmal1 in mESCs might be different from the typical transcriptional regulation exerted in coordination with Clock in other tissues because depletion of Bmal1 in mESCs leads to up-regulation of gene transcripts, suggesting Bmal1 is unexpectedly acting as a transcriptional repressor. Further studies will be needed to address this important aspect and clarify what is the molecular function of Bmal1 in the regulation of pluripotency.

Mice lacking Bmal1 from conception are viable and show a pleiotropic phenotype including loss of circadian rhythms, an acceleration of aging, and shortened life span (Bunger et al, 2000; Kondratov et al, 2006). Strikingly, a current report by the FitzGerald Lab shows that most of this phenotype is a consequence of the function of Bmal1 during embryo development (Yang et al, 2016). Our findings might help explain these interesting observations. *Bmal1−/−* mESCs do not show a severe block in cell differentiation, but instead they show milder defects that result in unbalanced regulation of lineage choice during cell differentiation. This type of defects during early embryo development might allow the formation of viable newborns in *Bmal1−/−* mice that accumulate widespread minor developmental defects that will cause tissue misfunctioning during adult life, in agreement with reported phenotypes in life span, fertility, body weight, blood glucose and arthropathy, atherosclerosis, and hair growth (Kondratov et al, 2006; Yang et al, 2016). In this context, it is important to note that given that the differentiation phenotype of *Bmal1−/−* mESCs is associated to altered cell signalling, it is likely that developmental defects observed in our in vitro differentiation systems are partially ameliorated during embryo development because of compensatory mechanism that take place in vivo and, thus, facilitate full gestation of *Bmal1−/−* mice.

In summary, our study provides solid proof that the core molecular clock component Bmal1 is expressed in pluripotent cells and that it regulates pluripotent cell differentiation. This suggests the existence of an uncharacterized function of the molecular clock that is essential for pluripotency execution and proper embryo development in mammals. Similar conclusions were drawn from an independently performed study, which recently came to our attention (Ameneiro et al, 2020).

# Materials and Methods

### mESC culture and differentiation

JM8 mESCs derived from the mouse strain C57BL6/N were cultured in primed serum conditions as previously described (Landeira et al,

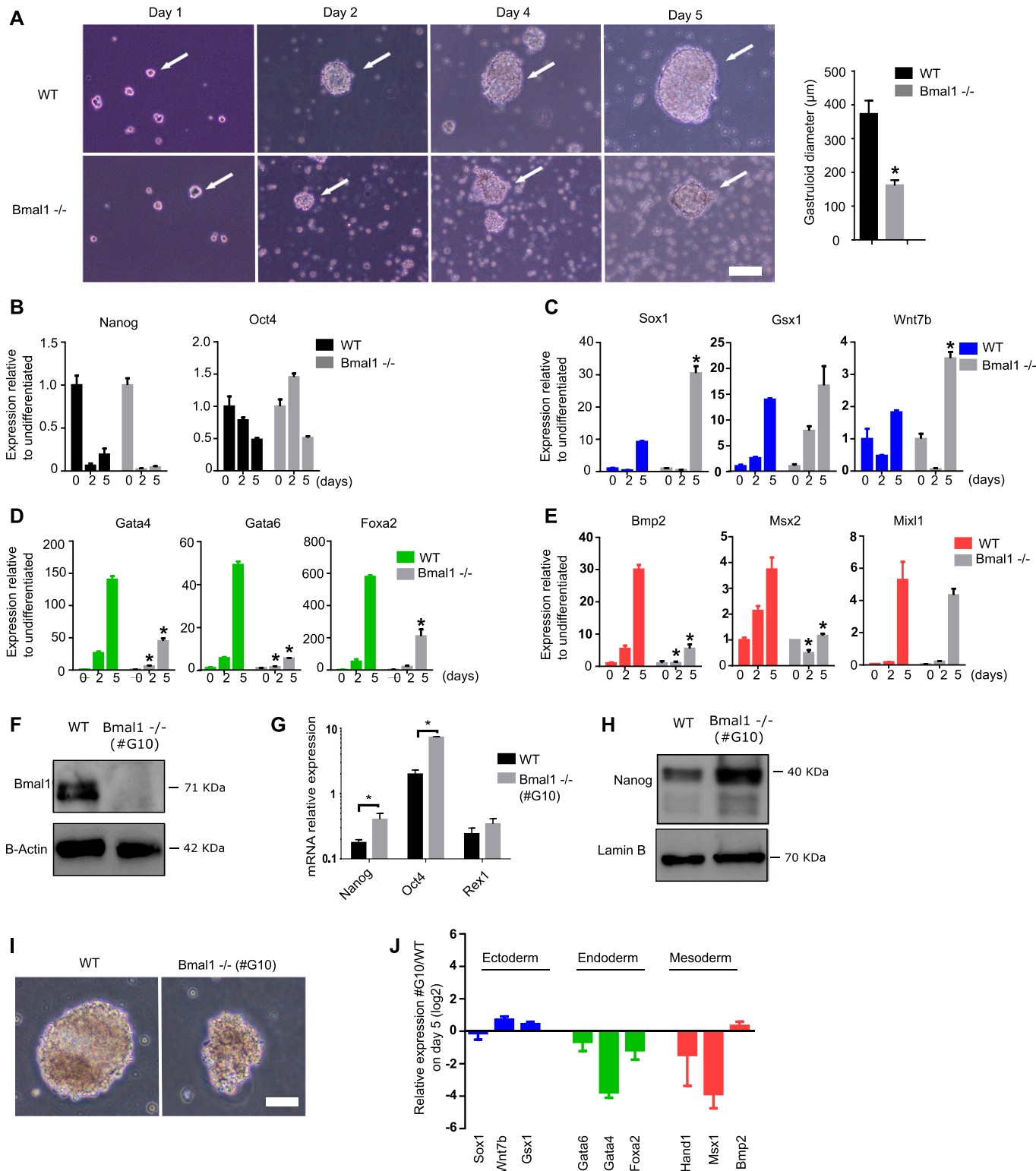

**Figure 5. *Bmal1−/−* mouse embryonic stem cells (mESCs) display altered formation of gastrula-like organoids.**
**(A)** Microscopic images of *Bmal1-/-* and JM8 wild-type mESCs during differentiation into gastruloids at indicated time points. White arrows indicate the position of the developing gastruloid. Scale bar is 50 μm. Histogram shows the average diameter of wild-type and *Bmal1-/-* gastruloids on day five. **(B, C, D, E)** mRNA expression of pluripotency (black)-, ectoderm (blue)-, endoderm (green)-, and mesoderm (red)-specific genes at indicated days during gastruloid formation as measure by RT-qPCR in JM8 wild-type and *Bmal1−/−* (grey) mESCs. Expression was normalized to *Hmbs* and *Ywhaz*. Fold change relative to undifferentiated cells at time cero is represented. **(F)** Western blot analysis of Bmal1 protein using whole-cell extracts of *Bmal1−/−* (clone #G10) and wildtype mESCs. B-actin provide a loading control. **(G)** mRNA expression

2015); DMEM KO (Gibco) media supplemented with 10% FCS (Gibco), LIF, penicillin/streptomycin (Gibco), L-glutamine (Gibco), and 2-mercaptoethanol (Gibco). Ground-state 2i mESCs were grown in serum-free 2i media as described previously (Ying et al, 2008): N2B27 supplemented with MEK inhibitor PD0325901 (1 $\mu$M) (Millipore) and GSK3 inhibitor CHIR99021 (3 $\mu$M) (Millipore) in the presence of LIF. Both serum-grown and 2i-grown mESCs were cultured feeder free on 0.1% gelatin-coated dishes at 37°C and 5% $CO_2$.

Neural stem cell cultures were derived from primed serum JM8 mESCs as previously described (Conti et al, 2005). Ground-state 2i JM8 and Bmal1−/− cells (1 × 10$^5$ cells in six well plates) were differentiated as a monolayer as described previously (Ying et al, 2003). JM8 and Bmal1−/− cells growing in serum (12,500 cells/cm$^2$) were differentiated by LIF removal as a monolayer as previously described (Landeira et al, 2010). Primed serum JM8 and Bmal1−/− cells (1 × 10$^4$ cells/ml) were induced to form gastrula-like organoids (gastruloids) as described elsewhere (Beccari et al, 2018).

## Teratoma formation and analysis

One million cells of each cell line were resuspended in 250 $\mu$l of PBS and injected subcutaneously into the flanks of NOD/SCID IL-2R$\gamma$−/− mice in the animal experimentation unit of the University of Granada. 5 wk later, the mice developed tumours that were removed, rinsed with PBS, fixed, and embedded in paraffin at the Biobank of Andalusia Public Health System. Tissue sections were cut and processed for haematoxylin and eosin staining or for immunohistochemistry staining with Anti-Human Smooth Muscle Actin (IR611; Dako), Ms Anti-Human Cytokeratin Clone AE1/AE3 (IR053; Dako), and Rb Anti-Human Glial Fibrillary Acidic Protein (IR 624; Dako). Species-specific secondary antibodies conjugated with peroxidase were used (EnVision Flex/HPR SM802; Dako). Immunohistochemistry staining images were quantified measuring the area that was positive for the signal of the antibody. For each staining, 10 images were quantified using Image J, averaged, and tested statistically using $t$ test.

## Derivation of *Bmal1−/−* mESCs

Single-guide RNA sequences (Oligo 1: 5′-CACCGAACCGGA-GAGTAGGTCGGT-3′, Oligo 2: 5′-AAACACCGACCTACTCTCCGGTTC-3′) were designed using CRISPR design tool and cloned into PX458 plasmid (Ran et al, 2013) upon testing with surveyor mutation detection kit (IDT). JM8 mESCs were transfected using lipofectamine 2000 (Thermo Fisher Scientific). Cells positive for GFP expression were sorted using flow cytometry and plated at serial dilutions to obtain isolated clonal colonies. DNA of 12 genetic clones was PCR-amplified (5′-TCTGCCTGGCTTATTCTTCC-3′ and 5′-AGTGCTGCTGGCCATTTAAG-3′) and sequenced upon pGEM-T (Promega) cloning. Four genetic clones showed mutations in both alleles and in the one containing mutation that produced a shorter version of Bmal1 protein was further characterized by RT-qPCR and Western blot.

## RT-qPCR and immunofluorescence

RNA was isolated using Trizol reagent (Thermo Fisher Scientific), reverse transcribed using RevertAid Frist Strand cDNA Synthesis kit (Thermo Fisher Scientific), and analysed by SYBRG real-time PCR using GoTaq qPCR Master Mix (Promega). Primers used are provided in Table S1.

Immunofluorescence analysis of mESCs colonies was carried out as described previously (Landeira et al, 2015). Briefly, the cells were fixed for 20 min in 2% paraformaldehyde, permeabilized 5 min in 0.4% Triton-X100, and blocked for 30 min in phosphate buffer saline supplemented with 0.05% Tween 20, 2.5% bovine serum albumin, and 10% goat serum. Primary antibodies used were Ms anti-Oct4 (611202; BD Biosciences), Rb anti-Nanog (RCAB002P-F; Cosmo Bio), and Ms anti-Nestin (MAB353; Merck). Secondary antibodies used were goat antimouse Alexa Fluor 488 (A-11001; Thermo Fisher Scientific) and goat antirabbit Alexa Fluor 555 (A-21429; Thermo Fisher Scientific). Slides were imaged by widefield fluorescence microscope Zeiss Axio Imager and images were analysed using Image J.

## Cell fractionation and Western blots

Separation of cytosol and nucleus fractions was carried out by resuspending 4 × 10$^6$ cells in 80 $\mu$l of cell lysis buffer, pH 7.9 (10 mM Hepes, 10 mM KCl, 100 $\mu$M EGTA, 100 $\mu$M EDTA, and freshly added 1 mM DTT, 1 mM PMSF, and protease inhibitors). After 15-min incubation on ice, 10 $\mu$l of cell lysis buffer with NP40 10% was added. Cytoplasmic supernatant was recovered by spinning at 16,200$g$, 4°C for 5 min. Pelleted nuclei were resuspended in 80 $\mu$l of nuclear lysis buffer (20 mM Hepes, 400 mM NaCl, 1 mM EGTA, 1 mM EDTA, and freshly added 1 mM DTT, 0.5 mM PMSF, and protease inhibitors). Soluble nuclear supernatant was recovered by spinning at 13,000 rpm, 4°C for 5 min. Cytoplasmic and nuclear fractions were mixed with equal volumes of 2× Laemmli loading buffer. In experiments where cell fractionation was not carried out, Western blots were carried out in whole-cell extracts lysed with 2× Laemmli loading buffer and using standard procedures. The following primary antibodies were used: Ms anti-Oct4 (611202; BD Biosciences), Rb anti-Nanog (RCAB002P-F; Cosmo Bio), Gt anti–Lamin B (SC-6216; Santa Cruz), Ms anti-ActB (A4700; Sigma-Aldrich), Rb anti-Bmal1 (ab93806; Abcam), Ms anti-MAP Kinase 2 (05-157; Merck), and Rb anti-Ezh2 (C15410039; Diagenode). Secondary species-specific antibodies conjugated to horseradish peroxidase were used: anti-rabbit-HRP

analysis of pluripotency markers *Nanog*, *Oct4*, and *Rex1* in Bmal1−/− (clone #G10) and JM8 wild-type mESCs by RT-qPCR. Expression was normalized to *Hmbs* and *Ywhaz*. **(H)** Western blot analysis of Nanog protein using whole-cell extracts of *Bmal1−/−* (clone #G10) and wild-type mESCs. Lamin B provide a loading control. **(I)** Microscopic images of gastruloids generated at day five by Bmal1−/− (clone #G10) and JM8 wild-type mESCs. Scale bar is 100 $\mu$m. **(J)** mRNA expression of ectoderm (blue)-, endoderm (green)-, and mesoderm (red)-specific genes on day five of gastruloid differentiation as measured by RT-qPCR in JM8 wild-type and Bmal1−/− (clone #G10) mESCs. Expression was normalized to Hmbs and Ywhaz. Fold change of Bmal1−/− #G10 relative to wild-type is represented. **(B, C, D, E, G, J)** Mean and SEM of three independent experiments are shown in (B, C, D, E, G, J). Asterisks (*) indicate significant difference (T-student $P < 0.05$).

(NA931; GE Healthcare), anti-mouse-HRP (NA931; GE Healthcare), and anti-goat-HRP (ab97110; Abcam). Clarity Western ECL reagents (Bio-Rad) was used for detection.

## mRNA-seq and bioinformatic analysis

Total RNA of JM8 wild-type and Bmal1–/– mESCs growing in the 2i + LIF medium was isolated using Trizol (Thermo Fisher Scientific), and mRNA library preparation was carried out using TruSeq Stranded mRNA kit (Illumina). 30 million 75-bp paired end reads were analysed per sample. Two biological replicates were carried out for each cell line. Library preparation and Illumina sequencing was carried out at National Center for Genomic Analysis-Center for Genomic Regulation (CNAG-CRG). Raw FASTQ files were quality filtered and aligned to the mouse genome (version mm9) using STAR 2.5.2 (Dobin & Gingeras, 2015). Gene expression was normalized with trimmed mean of M values (Robinson & Oshlack, 2010) implemented in NOISeq R package (Tarazona et al, 2015). Differential expression analysis was performed with DESeq2 package (Love et al, 2014). MA plot was generated with edgeR (McCarthy et al, 2012) and heat maps were generated with pheatmap package. mRNA of deregulated genes (log2 fold change greater than 1 or smaller than –1 and adjusted $P$-value < 0.05) were used as input gene lists in DAVID bioinformatic gene enrichment analysis. List of differentially expressed genes is provided in Table S1.

Single-cell RNA expression of Bmal1 and Oct4 in developing blastocysts was extracted from published data (Deng et al, 2014). The cells were classified by the level of expression of Oct4 in four quartile groups (Oct4 low correspond to bottom quartile and Oct4 high corresponds to top quartile), and average expression of Bmal1 was plotted.

## Data access

Datasets are available at GEO-NCBI with accession number GSE134884 (https://www.ncbi.nlm.nih.gov/geo/query/acc.cgi?acc=GSE134884).

# Supplementary Information

# Acknowledgements

We are grateful to Maria Romo and Maria Quintana for technical support. We thank core facilities in GENYO and in particular to the genomics and bioinformatics units. We also thank the genomics unit at the National Center for Genomic Analysis-Center for Genomic Regulation (CNAG-CRG) for assistance with mRNA-seq experiments. D Landeira was supported by a Ramón y Cajal grant of the Spanish ministry of economy and competitiveness (RYC-2012-10019). The Landeira lab is supported by the Spanish ministry of economy and competitiveness (BFU2016-75233-P) and the Andalusian regional government (PC-0246-2017). R Montes was supported by an Fundación Progreso y Salud (FPS) contract, and the Ramos-Mejia lab is supported by Instituto de Salud Carlos III-FEDER grants (CPII17/00032 and PI17/01574). L Lopez-Onieva is supported by a reincorporation grant from the University of Granada 2017 (Programa de incorporación de doctores del plan propio).

## Author Contributions

A Gallardo: data curation, formal analysis, and investigation.
A Molina: investigation.
HG Asenjo: investigation.
J Martorell-Marugán: investigation.
R Montes: investigation and methodology.
V Ramos-Mejia: methodology.
A Sanchez-Pozo: methodology.
P Carmona-Sáez: methodology.
L Lopez-Onieva: methodology and funding acquisition.
D Landeira: conceptualization, investigation, and funding acquistion.

## Conflict of Interest Statement

The authors declare that they have no conflict of interest.

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
