## [Reviewer comments · Life Science Alliance]

Life Science Alliance

The molecular clock protein Bmal1 regulates cell differentiation in mouse embryonic stem cells

Amador Gallardo, Aldara Molina, Helena Asenjo, Jordi Martorell-Marugán, Rosa Montes, Veronica Ramos-Mejia, Antonio Sanchez-Pozo, Pedro Carmona-Sáez, and David Landeira

DOI: <https://doi.org/10.26508/lsa.201900535>

Corresponding author(s): David Landeira, Centre for genomics and oncological research (GENYO)

Review Timeline:

Submission Date:	2019-08-25
Editorial Decision:	2019-10-16
Appeal Received:	2019-10-17
Editorial Decision:	2019-10-23
Revision Received:	2020-02-24
Editorial Decision:	2020-03-10
Revision Received:	2020-03-12
Accepted:	2020-03-23

Scientific Editor: Andrea Leibfried

Transaction Report:

October 16, 2019

Re: Life Science Alliance manuscript #LSA-2019-00535-T

Dr. David Landeira
Centre for genomics and oncological research (GENYO)
Av. de la Ilustración 114
Granada 18016
Spain

Dear Dr. Landeira,

Thank you for submitting your manuscript entitled "The molecular clock protein Bmal1 regulates cell differentiation in pluripotent mouse embryonic stem cells" to Life Science Alliance. Please excuse the delay in getting back to you, we had to give the reviewers more time in this case. We have now heard back from two reviewers on your work and their reports are attached below.

As you will see, reviewer #1 points out that many of your conclusions are not sufficiently supported by the data provided. This reviewer also thinks that the Bmal1 antibody used is unspecific and s/he notes other data quality issues. Both reviewers furthermore would have expected further analyses such as replication studies, additional lines of evidence for the activation of mesoderm genes during lineage transition as well as teratoma assays.

Given these concerns, we are afraid we do not have the level of reviewer support that we would need to proceed further with the paper. We are thus returning your manuscript to you with the message that we cannot publish it here.

We are sorry our decision is not more positive, but hope that you find the reviews constructive. Of course, this decision does not imply any lack of interest in your work and we look forward to future submissions from your lab.

Thank you for your interest in Life Science Alliance.

Sincerely,

Andrea Leibfried, PhD
Executive Editor
Life Science Alliance

Reviewer #1 (Comments to the Authors (Required)):

In this manuscript, the authors studied the functions of Bmal1 in regulating cell differentiation. Using RNA-seq analysis, they analyzed that Bmal1^{-/-} mESCs showed altered expression of key pluripotency-associated signaling pathways. They claimed that although Bmal1 was expressed in

mESCs, but was not essential for mESC self-renewal. Deletion of Bmal1 resulted in deficient cell differentiation during the formation of gastrula-like organoids.

Although the paper presents some findings, the results are still limited and more evidences are needed to support the conclusions. Additionally, the underlining mechanism of Bmal1 in regulating cell differentiation is still elusive and further experiments should be performed. Lastly, the quality of figures should be further improved.

Major concerns:

- (1) The antibody of Bmal1 seems not specific, which detects several bands in western blotting. The patterns of Bmal1 in different figures are different, eg., Fig. 1G, 1J and 2E. The authors should explain the reasons. In addition, statistical analysis of western blotting is needed.
- (2) Statistical analysis of western blotting and fluorescence intensity analysis of Nanog in Fig. 2G and 2H is required.
- (3) In Figure 3G, the author demonstrated that MAPK pathway might underline the mechanism of the upregulation of Nanog protein observed in Bmal1 ^{-/-} mESCs. The authors need to provide more experimental evidences to support the conclusion.
- (4) Although the authors analyzed pluripotency exit and lineage transition using in vitro differentiation systems, in vivo teratoma formation should be performed to check whether BMAL1 KO ESCs can form the three germ layers.
- (5) To draw the conclusion that deletion of Bmal1 affects the activation of mesoderm genes during lineage transition, more markers should be detected. Besides the differentiation of gastrula-like organoids, other mesoderm cells or organoids should be differentiated.
- (6) Many typos were found in the manuscript, and the improvement of writing is needed. For example, in results part, paragraph 3, line 5, "these results solidly demonstrate that albeit mESCs do now display circadian oscillations", here "now" should be "not", otherwise, the context will be very confusing.

Reviewer #3 (Comments to the Authors (Required)):

Overall, this manuscript demonstrates that even though mESCs do not show oscillation of clock genes, the clock protein Bmal1 is expressed and plays a role in regulating ES cell differentiation. In particular, expression of mesoderm markers is impaired in the absence of Bmal1. The study is descriptive and does not identify the underlying mechanisms, but this is acceptable for LSA. The conclusions are appropriate for the data presented. My primary concern is it appears that the conclusions are based on analysis of a single Bmal1^{-/-} clonal line. I recommend re-expressing Bmal1 in the KO cells to show that the effects on differentiation are specifically due to lack of Bmal1 and not a clonal effect. At minimum, authors should validate their findings in the other Bmal1^{-/-} clones that they generated.

Dear Dr. Leibfried,

Thanks for your email and thanks for taking the time to revise our manuscript.

I am writing because I would like you to reconsider your decision of rejecting our study and not giving us the opportunity to resubmit a revised version addressing reviewers' concerns. Upon careful examination of the reviewer's comments, I feel that both referees were pretty supportive. They did not ask for any major experiment that might alter the main message of the manuscript nor they asked for any technically demanding assay. In fact, they only asked for minor revisions that we can easily address.

I am attaching a point-by-point rebuttal letter explaining how we would address reviewers' concerns. I would be grateful if you could take the time to consider whether those amendments might be satisfactory enough to send the manuscript for a second round of review. Please note that among the seven raised issues (six by referee #1 and one by referee #3), we have already solved three of them and have the reagents and technical platforms to tackle the remaining four within six weeks.

Please, let us know if you are interested in considering a revised version of our manuscript.

Best wishes,

David Landeira

We thank to referees #1 and #3 for their positive feedback and constructive comments that will improve our manuscript making it more complete and solid. To facilitate the review process, we have used blue colour font for our comments in this rebuttal letter.

Reviewer #1 (Comments to the Authors (Required)):

In this manuscript, the authors studied the functions of Bmal1 in regulating cell differentiation. Using RNA-seq analysis, they analyzed that Bmal1^{-/-} mESCs showed altered expression of key pluripotency-associated signaling pathways. They claimed that although Bmal1 was expressed in mESCs, but was not essential for mESC self-renewal. Deletion of Bmal1 resulted in deficient cell differentiation during the formation of gastrula-like organoids.

Although the paper presents some findings, the results are still limited and more evidences are needed to support the conclusions. Additionally, the underlining mechanism of Bmal1 in regulating cell differentiation is still elusive and further experiments should be performed. Lastly, the quality of figures should be further improved.

We thank the reviewer for appreciating the interest of our work.

Major concerns:

(1) The antibody of Bmal1 seems not specific, which detects several bands in western blotting. The patterns of Bmal1 in different figures are different, eg., Fig. 1G, 1J and 2E. The authors should explain the reasons.

We apologize for not having carefully explained this in our previous version of the manuscript. The anti-Bmal1 antibody used in this manuscript (abcam) detects a prominent band of around 70 kDa and two more slightly heavier secondary bands (Fig. 1G and 1J). These bands can also be detected using other anti-Bmal1 antibodies and correspond to different phosphorylation states of Bmal1 protein (Sahar et al., 2010; Yoshitane et al., 2009). Importantly, the signal of all three bands is lost in Bmal1^{-/-} mESCs, confirming that our anti-Bmal1 antibody is specific and recognizes different phosphorylated versions of Bmal1 protein (Fig. 2E). We will discuss this in the main text of our manuscript to avoid misunderstanding.

In addition, statistical analysis of western blotting is needed.

We will include statistical analysis for westerns in 1E, 1G and 1J:

(2) Statistical analysis of western blotting and fluorescence intensity analysis of Nanog in Fig. 2G and 2H is required.

We will include statistical analysis of experiments showed in Fig. 2G and 2H as requested by the referee:

(3) In Figure 3G, the author demonstrated that MAPK pathway might underline the mechanism of the upregulation of Nanog protein observed in *Bmal1*^{-/-} mESCs. The authors need to provide more experimental evidences to support the conclusion.

RNA-seq analysis of *Bmal1*^{-/-} mESCs suggest deregulation of MAPK signalling pathway in these cells. MAPK pathway is a well-established regulator of Nanog expression. To support our conclusion we will compare expression of MAPK components by RT-qPCR/Western blots in wt and *Bmal1*^{-/-} mESCs.

(4) Although the authors analyzed pluripotency exit and lineage transition using in vitro differentiation systems, in vivo teratoma formation should be performed to check whether BMAL1 KO ESCs can form the three germ layers.

Our animal facility will analyse the potential to form teratomas of wt and *Bmal1*^{-/-} mESCs and compare the formation of the three germ layers.

(5) To draw the conclusion that deletion of *Bmal1* affects the activation of mesoderm genes during lineage transition, more markers should be detected. Besides the differentiation of gastrula-like organoids, other mesoderm cells or organoids should be differentiated.

We will design primers to amplify other mesoderm markers and analyse the expression of these genes in three differentiation systems: serum monolayer, 2i monolayer and gastruloid formation. These experiments will also be complemented by the analysis of mesoderm formation during in vivo teratoma assays described in (5).

(6) Many typos were found in the manuscript, and the improvement of writing is needed. For example, in results part, paragraph 3, line 5, "these results solidly demonstrate that albeit mESCs do now display circadian oscillations", here "now" should be "not", otherwise, the context will be very confusing.

Typos will be corrected.

Reviewer #3 (Comments to the Authors (Required)):

We thank the reviewer for appreciating that the conclusions are appropriate for the data presented.

Overall, this manuscript demonstrates that even though mESCs do not show oscillation of clock genes, the clock protein Bmal1 is expressed and plays a role in regulating ES cell differentiation. In particular, expression of mesoderm markers is impaired in the absence of Bmal1. The study is descriptive and does not identify the underlying mechanisms, but this is acceptable for LSA. The conclusions are appropriate for the data presented. My primary concern is it appears that the conclusions are based on analysis of a single Bmal1^{-/-} clonal line. I recommend re-expressing Bmal1 in the KO cells to show that the effects on differentiation are specifically due to lack of Bmal1 and not a clonal effect. At minimum, authors should validate their findings in the other Bmal1^{-/-} clones that they generated.

We will use Bmal1^{-/-} mESCs independently derived and characterized currently available in my lab to confirm that the lack of Bmal1 leads to upregulation of Nanog protein (western blot), defective formation of gastruloids and upregulation of mesoderm markers.

Bibliography

- Sahar, S., Zocchi, L., Kinoshita, C., Borrelli, E., and Sassone-Corsi, P. (2010). Regulation of BMAL1 protein stability and circadian function by GSK3beta-mediated phosphorylation. *PloS one* 5, e8561.
- Yoshitane, H., Takao, T., Satomi, Y., Du, N.H., Okano, T., and Fukada, Y. (2009). Roles of CLOCK phosphorylation in suppression of E-box-dependent transcription. *Molecular and cellular biology* 29, 3675-3686.

MS: LSA-2019-00535-T

Dr. David Landeira
Centre for genomics and oncological research (GENYO)
Av. de la Ilustración 114
Granada 18016
Spain

Dear Dr. Landeira,

Thank you for your recent correspondence regarding our decision on your manuscript "The molecular clock protein Bmal1 regulates cell differentiation in pluripotent mouse embryonic stem cells".

We appreciate your revision plan and your willingness to extend the analysis significantly. We therefore decided that it is warranted to consider such a revised version. Please note, however, that we would need strong support from reviewer #1 on the revised version. It may be good to address the specific concern regarding the Bmal1 antibody in a different way, as quantifications and statistical analyses of the blots already at hand may get confounded by the multiple bands observed.

Sincerely,

Andrea Leibfried, PhD
Executive Editor
Life Science Alliance

We thank to referees #1 and #3 for their positive feedback and constructive comments that have improved our manuscript making it more complete and solid. To facilitate the review process, we have used blue font for new major changes of text in the manuscript as well as for our comments in this rebuttal letter.

Reviewer #1 (Comments to the Authors (Required)):

In this manuscript, the authors studied the functions of Bmal1 in regulating cell differentiation. Using RNA-seq analysis, they analysed that Bmal1^{-/-} mESCs showed altered expression of key pluripotency-associated signalling pathways. They claimed that although Bmal1 was expressed in mESCs, but was not essential for mESC self-renewal. Deletion of Bmal1 resulted in deficient cell differentiation during the formation of gastrula-like organoids.

Although the paper presents some findings, the results are still limited and more evidences are needed to support the conclusions. Additionally, the underlining mechanism of Bmal1 in regulating cell differentiation is still elusive and further experiments should be performed. Lastly, the quality of figures should be further improved.

We thank the reviewer for appreciating the interest of our work.

Major concerns:

(1) The antibody of Bmal1 seems not specific, which detects several bands in western blotting. The patterns of Bmal1 in different figures are different, e.g., Fig. 1G, 1J and 2E. The authors should explain the reasons.

We apologize for not having carefully explained this in our previous version of the manuscript. The anti-Bmal1 antibody used in this manuscript (abcam ab93806) detects a prominent band of around 70 kDa and two more slightly heavier secondary bands (Fig. 1E, G and 1J). These bands can also be detected using other anti-Bmal1 antibodies and correspond to different phosphorylation states of Bmal1 protein (Sahar et al., 2010; Yoshitane et al., 2009). Importantly, the signal of all three bands is lost in our Bmal1^{-/-} mESCs (the one included in the previous version and newly derived cell line), demonstrating that our abcam anti-Bmal1 antibody is specific and recognizes different phosphorylated versions of Bmal1 protein (Fig. 2E and 5F). In fitting, this antibody has been extensively used in other studies (more than 31 reports):

<https://www.abcam.com/top-555.5556030273438>

We have discussed this in the main text of our manuscript to avoid misunderstanding (results section, page 4).

In addition, statistical analysis of western blotting is needed.

As requested, in new figure S1A, B and C we have included statistical analysis for westerns in 1E, 1G and 1J respectively.

(2) Statistical analysis of western blotting and fluorescence intensity analysis of Nanog in Fig. 2G and 2H is required.

We have now included statistical analysis of experiments showed in Fig. 2G (western blot) and 2H (immunofluorescence) as requested by the referee. See quantification of Wester-blot in Fig. S1D and statistical analysis of immunofluorescence in Fig. 2I

(3) In Figure 3G, the author demonstrated that MAPK pathway might underline the mechanism of the upregulation of Nanog protein observed in Bmal1^{-/-} mESCs. The authors need to provide more experimental evidences to support the conclusion.

We have looked into the possibility that MAPK/Erk signalling was responsible of Nanog protein upregulation found in Bmal1^{-/-} mESCs. However, this does not seem to be case because expression of Erk1/2 protein was unaltered in Bmal1^{-/-} mESCs growing in both primed serum and naïve 2i

conditions (Figure S2C), and mRNA expression of Erk1/2 targets was not significantly affected by Bmal1 depletion (Figure S2D and E), suggesting that upregulation of Nanog protein is not a consequence of reduced Erk1/2 signalling in *Bmal1*^{-/-} mESCs. In addition, the fact that *Bmal1*^{-/-} mESCs show augmented level of Nanog when grown in the presence of the Erk inhibitor (2i media) (Fig. 2G) also supports that Nanog up-regulation is Erk-independent. We have included these experiments in the manuscript and commented them accordingly on page 7.

(4) Although the authors analyzed pluripotency exit and lineage transition using in vitro differentiation systems, in vivo teratoma formation should be performed to check whether BMAL1 KO ESCs can form the three germ layers.

We thank the referee for suggesting this experiment. We have compared wild-type and *Bmal1*^{-/-} mESCs during teratoma formation. We have carried out qualitative and quantitative analysis by haematoxylin/eosin and immunohistochemistry respectively and found that *Bmal1*^{-/-} mESCs can differentiate into the three germ layers albeit with reduced efficiency compared to wild-type cells. This result is in fitting with the reduced differentiation efficiency of *Bmal1*^{-/-} mESCs during in vitro differentiation that we found and with previously reported full gestation of *Bmal1*^{-/-} mouse embryos. Teratoma analyses are included in Fig. 3G, H, I and commented in results section on page 7.

(5) To draw the conclusion that deletion of Bmal1 affects the activation of mesoderm genes during lineage transition, more markers should be detected. Besides the differentiation of gastrula-like organoids, other mesoderm cells or organoids should be differentiated.

We agree with the referee that the study would be benefit from a more detailed analysis of the differentiation phenotype of *Bmal1*^{-/-} mESCs. We have increased the number of ectoderm, endoderm and mesoderm genes analysed in all the differentiation experiments included in the manuscript: serum monolayer, 2i monolayer and gastruloid formation. We consistently observe reduced activation of endoderm and mesoderm marker genes in *Bmal1*^{-/-} mESCs across all systems (Figure 4, 5, S3 and S4). In addition, analysis of teratoma formation in *Bmal1*^{-/-} mESCs (Figure 3G, H and I) and our new analysis of gastruloid formation by the newly derived *Bmal1* mutant cell line (*Bmal1*^{-/-} clone #G10) (Fig. I, J, S4C and S4D) further demonstrate the requirement of Bmal1 for multi lineage differentiation of pluripotent cells.

(6) Many typos were found in the manuscript, and the improvement of writing is needed. For example, in results part, paragraph 3, line 5, "these results solidly demonstrate that albeit mESCs do now display circadian oscillations", here "now" should be "not", otherwise, the context will be very confusing.

Typos have been corrected.

Reviewer #3 (Comments to the Authors (Required)):

We thank the reviewer for appreciating the interest of our study that the conclusions are appropriate for the data presented.

Overall, this manuscript demonstrates that even though mESCs do not show oscillation of clock genes, the clock protein Bmal1 is expressed and plays a role in regulating ES cell differentiation. In particular, expression of mesoderm markers is impaired in the absence of Bmal1. The study is descriptive and does not identify the underlying mechanisms, but this is acceptable for LSA. The conclusions are appropriate for the data presented. My primary concern is it appears that the conclusions are based on analysis of a single Bmal1^{-/-} clonal line. I recommend re-expressing Bmal in the KO cells to show that the effects on differentiation are specifically due to lack of Bmal1 and not a clonal effect. At minimum, authors should validate their findings in the other Bmal1^{-/-} clones that they generated.

We thank the referee for this comment. We have now derived a new clonal Bmal1 knockout mESC line (clone #G10) (Fig. 5F, G, S4A, S4B) and confirmed that the phenotype described for previous *Bmal1*^{-/-} mESCs is a consequence of the depletion of Bmal1 protein. In particular, we have confirmed that depletion of Bmal1 protein in *Bmal1*^{-/-} (clone #G10) mESCs also leads to higher level of Nanog protein (Fig. 5H) and smaller gastruloids (Fig 5I, S4C) in which expression of endoderm and mesoderm

differentiation markers is hindered (Fig. 5J and S4D). Results are described in page 9 of the new manuscript.

Bibliography

Sahar, S., Zocchi, L., Kinoshita, C., Borrelli, E., and Sassone-Corsi, P. (2010). Regulation of BMAL1 protein stability and circadian function by GSK3beta-mediated phosphorylation. *PloS one* 5, e8561.

Yoshitane, H., Takao, T., Satomi, Y., Du, N.H., Okano, T., and Fukada, Y. (2009). Roles of CLOCK phosphorylation in suppression of E-box-dependent transcription. *Molecular and cellular biology* 29, 3675-3686.

March 10, 2020

RE: Life Science Alliance Manuscript #LSA-2019-00535-TR-A

Dr. David Landeira
Centre for genomics and oncological research (GENYO)
Av. de la Ilustración 114
Granada, Granada 18016
Spain

Dear Dr. Landeira,

Thank you for submitting your revised manuscript entitled "The molecular clock protein Bmal1 regulates cell differentiation in mouse embryonic stem cells". As you will see, reviewer #1 appreciates the changes introduced in revision, and we would thus be happy to publish your paper in Life Science Alliance pending final revisions necessary to meet our formatting guidelines:

- Please make sure that the author order in our submission system and the one in your manuscript file match
- Please provide your manuscript text as a docx file

A. FINAL FILES:

-- Summary blurb (enter in submission system): A short text summarizing in a single sentence the study (max. 200 characters including spaces). This text is used in conjunction with the titles of papers, hence should be informative and complementary to the title. It should describe the context and significance of the findings for a general readership; it should be written in the present tense

and refer to the work in the third person. Author names should not be mentioned.

B. MANUSCRIPT ORGANIZATION AND FORMATTING:

Sincerely,

Reviewer #1 (Comments to the Authors (Required)):

This revised version has addressed some of major concerns and the authors have performed a number of additional assays in response to suggestions or criticisms by the reviewers. Indeed, the data in the new version provide better support for the authors' claims that Bmal1^{-/-} mESCs showed altered expression of key pluripotency associated signalling pathways.

March 23, 2020

RE: Life Science Alliance Manuscript #LSA-2019-00535-TRR

Dr. David Landeira
Centre for genomics and oncological research (GENYO)
Av. de la Ilustración 114
Granada, Granada 18016
Spain

Dear Dr. Landeira,

Thank you for submitting your Research Article entitled "The molecular clock protein Bmal1 regulates cell differentiation in mouse embryonic stem cells". It is a pleasure to let you know that your manuscript is now accepted for publication in Life Science Alliance. Congratulations on this interesting work.

DISTRIBUTION OF MATERIALS:

Again, congratulations on a very nice paper. I hope you found the review process to be constructive and are pleased with how the manuscript was handled editorially. We look forward to future exciting submissions from your lab.

Sincerely,
